# Anti-Inflammatory Effects of Polyphenols from Plum (*Prunus salicina Lindl*) on RAW264.7 Macrophages Induced by Monosodium Urate and Potential Mechanisms

**DOI:** 10.3390/foods12020254

**Published:** 2023-01-05

**Authors:** Yibin Li, Wei Deng, Li Wu, Shouhui Chen, Zhipeng Zheng, Hongbo Song

**Affiliations:** 1College of Food Science, Fujian Agriculture and Forestry University, Fuzhou 350002, China; 2Research Institute of Agri-Engineering and Technology, Fujian Academy of Agricultural Sciences, Fuzhou 350003, China; 3Key Laboratory of Subtropical Characteristic Fruits, Vegetables and Edible Fungi Processing, Ministry of Agriculture and Rural Affairs, Fuzhou 350003, China

**Keywords:** *Prunus salicina Lindl*, polyphenols, RNA-seq, pathways

## Abstract

Acute gouty arthritis is an acute inflammatory reaction caused by the deposition of monosodium urate (MSU) crystals in joints and surrounding soft tissues. Controlling inflammation is the key to preventing acute gouty arthritis. Anti-inflammatory activities and the possible molecular mechanisms of plum (*Prunus salicina Lindl* cv. “furong”) polyphenols (PSLP) on RAW264.7 macrophage cells induced by monosodium urate were investigated. PPSF significantly inhibited the activity of inflammatory factors such as tumor necrosis factor-α (TNF-α), interleukin-1β (IL-1β), and interleukin-18 (IL-18). In addition, PPSF exhibited excellent activation of superoxide dismutase (SOD) activity and reduction of intracellular reactive oxygen species (ROS) and malondialdehyde (MDA) levels in RAW264.7 macrophages. The results of global screening of all transcripts by RNA-seq revealed 8585 differentially expressed genes between the PSLP-treated group and the MUS group. From GO analysis, PSLP could affect the occurrence and development of RAW264.7 macrophage inflammation through biological processes, such as organic substance metabolism, intracellular organelles, and binding function. The regulation mechanism of PSLP on MSU-induced RAW264.7 macrophage inflammation may be achieved through the HIF-1 signaling pathway, renal cell carcinoma, the ErbB signaling pathway, and the FoxO signaling pathway. Therefore, PSLP has great prospects in the prevention of gout and similar inflammatory diseases.

## 1. Introduction

Acute gouty arthritis (AGA), a common inflammatory reaction, is caused by elevated blood uric acid levels and the deposition of monosodium urate (MSU) crystals in joints and soft tissues [1]. Macrophage phagocytosis of MSU is the core of acute gouty arthritis [2]. At present, anti-inflammatory drugs, including nonsteroidal anti-inflammatory drugs, corticosteroids, and colchicine, are mainly used to prevent and treat gout flares. However, these drugs are associated with many adverse effects, such as liver damage, kidney damage, and allergic reactions, which may limit their clinical application [3]. Therefore, the search for better anti-inflammatory and low-toxic drugs for the treatment of acute gouty arthritis is a necessity.

Natural polyphenols act as an anti-inflammatory modulator, and their anti-inflammatory mechanism is through the recognition of macrophage surface receptors, which activates macrophages and increases pro-inflammatory effectors, including tumor necrosis factor α (TNF-α), interleukin-6 (IL-6) and interleukin-1β (IL-1β) [4]. Resveratrol had anti-inflammatory effects on MSU crystal-induced inflammation in vitro and in vivo, including reducing pro-inflammatory cytokines, inflammatory cell recruitment, and foot swelling [5]. Cheng [6] studied the anti-inflammatory effects of blueberry polyphenols on LPS-induced RAW264.7 cells and showed that blueberry polyphenols could exert anti-inflammatory effects by inhibiting the expression of inflammatory cytokines IL-1, IL-6, and IL-12. Arbizu-berrocal [7] found that mango polyphenols significantly reduced TNF-α-stimulated NF-κB expression and phosphorylation in non-cancerous MCF-12A cells and reduced inflammation in non-cancerous tissues by up-regulating mRNA and protein expression levels of miR-126. Du [8] found that pomegranate peel polyphenols could inhibit the release and expression of pro-inflammatory cytokines TNF-α, IL-1β, and IL-6, thus showing good anti-inflammatory effects in the LPS-induced RAW264.7 macrophage inflammation model. Prasanna [9] found that onion lectin could activate macrophages and promote macrophage pro-inflammatory responsiveness, which eventually facilitates the Th1 immune response in T lymphocytes. In addition, grape polyphenols significantly reduced the secretion of pro-inflammatory cytokines IL-1, IL-6, IL-12, and TNF-α levels by activating human peripheral blood leukocytes, showing anti-inflammatory effects [10]. Plum (*Prunus salicina Lindl* cv. “furong”) is widely planted in southern China, such as in Fujian Province. Polyphenol extracts of plum (PSLP) had an excellent ability to inhibit xanthine oxidase and scavenge free radicals’ activity [11,12]. However, there is little literature on the anti-inflammatory effect of PSLP. In the current study, we study whether PSLP can inhibit MSU crystal-induced inflammation.

Based on the above information, the purpose of our experiment was to investigate the anti-inflammatory activities of PSLP in RAW264.7 cells induced by monosodium urate. Its effects were evaluated from the perspectives of inhibition of inflammatory factor secretion and regulation of related gene expression. In addition, the underlying molecular mechanisms using RNA-Seq were explored.

## 2. Materials and Methods

### 2.1. Materials and Reagents

*Prunus salicina Lindl* was purchased from Yongtai County, Fuzhou City. RAW264.7 cells, a murine monocyte-macrophage cell line, were obtained from the cell bank of the Institute of Chinese Academy of Sciences (Shanghai, China). Dulbecco’s modified Eagle’s medium (DMEM), phosphate-buffered saline (PBS), Fetal bovine serum (FBS), and Hank’s balanced salt solution (HBSS) were purchased from Thermo Fisher Technology (China) Co. Ltd. (Shanghai, China). The cell counting kit-8 (CCK-8) was available from Shanghai Ya Mei Biopharmaceutical Technology Co. Ltd. (Shanghai, China); the TNF-α, IL-6, IL-1β, and IL-8 ELISA kits were purchased from Wuhan Hua Mei Biological Engineering Co. Ltd. (Wuhan, China). The SOD and MDA assay kits were obtained from the Nanjing Jian Cheng Institute of Biological Engineering (Nanjing, China). The RNA reverse transcription kit and RT-PCR quantitative kit were purchased from Tian Gen Biochemical Technology Co, Ltd. (Beijing, China).

### 2.2. Reagent Preparation Method

#### 2.2.1. Preparation of Polyphenols

The extraction method of plum polyphenols was employed as previously described [13,14,15]. The polyphenols were extracted in an ultrasonic bath (KQ-600DV, 40 kHz, 420 W, Kunshan Ultrasonic Instrument Co., Jiangsu, China). The extraction parameters were as follows: the extraction solvent was 60% ethanol, the solid-liquid ratio was 1:20, the ultrasonic power was 420 W, the extraction temperature was 50 °C, and the ultrasonic time was 45 min. After extraction, the mixture was centrifuged at 3000 rpm for 15 min. The supernatant was condensed to 17% solid content by a vacuum rotary evaporator (Senco-GG17, Shanghai Shenke Technology Co., Ltd., Shanghai, China) at 0.09 MPa and 45 °C. The concentrated solution was purified by macroporous resin, and the purified solution was freeze-dried to obtain plum polyphenols. The total phenol content of the extract was 680.97 ± 31.05 mg GAE/g. High-performance liquid chromatography was used to identify eight polyphenols in PSLP, including flavan-3-ols (1.383 mg/kg proantho cyanidins B1, 0.048 mg/kg proantho cyanidins B2, 1.068 mg/kg catechin, and 5.596 mg/kg epicatechin), flavonols (0.549 mg/kg myricetin and 1.097mg/kg quercetin), and phenolic acids (1.123 mg/kg chlorogenic acid, 1.162 mg/kg protocatechuic acid, and 0.559 mg/kg ferulic acid).

#### 2.2.2. Preparation of MSU

MSU crystals were prepared according to Li [5] et al. with some modifications. Briefly, 1 g of uric acid was dissolved in 200 mL of distilled water. Subsequently, 6 mL of 1 M NaOH was added, followed by heating to 100 °C. The pH of the solution was adjusted to about 7.2 with HCl, centrifuged for 3 min (8000 r/min), and then stored in physiological saline. MSU was sterilized by autoclaving and then used for cell experiments. A small amount of MSU was dried in a drying oven at 180 °C to a constant weight, and then 100 mg of dried MSU was weighed and added to 10 mL of DMEM culture solution to prepare MSU solution with a mass concentration of 10 mg/mL.

### 2.3. Cell Culture

RAW264.7 cells, a murine monocyte-macrophage cell line, were maintained in DMEM supplemented with 10% fetal bovine serum, 100 U/mL penicillin, and 100 U/mL streptomycin. In addition, the RAW264.7 were incubated at 37 °C in a humidified incubator at 5% CO_2_ for the indicated time. The experiment was divided into five groups, including a blank control group (control, an equal volume of the solvent used to prepare the test substances replaced them), the MSU model group (MSU, treated with 100 μL of 2 mg/mL MSU solution for 24 h), the 20 μg/mL polyphenol group, the 40 μg/mL polyphenol group, the 60 μg/mL polyphenol group (PSLP, all the PSLP-treated groups were treated with 2 mg/mL MSU solution for 24 h and then were combined with 100 μL of a different concentration of a PSLP solution for 24 h), the positive control group (Col, treated with 100 μL of 2 mg/mL MSU solution for 24 h and 100 μL of a 0.4 μg/mL colchicine solution was added for 24 h). Each group was repeated five times.

### 2.4. Determination of Cell Viability

The cell viability of PSLP treatment on RAW264.7 cells was determined by a cell-counting kit-8 (CCK-8) assay, according to the method described by Wu [16] with slight modifications. The number of RAW264.7 cells was adjusted to 5 × 10^5^ cells/mL with DMEM containing 10% fetal bovine serum and inoculated in 96-well plates with 100 μL per well, and three replicate wells after the cells were plastered with different concentrations of polyphenols for 24 and 48 h. The supernatant was discarded, and 100 μL of DMEM containing 10 μL CCK-8 solution was added to each well and incubated for 2 h in the incubator. The absorbance was obtained with a spectrophotometer at 540 nm. Cytotoxicity was expressed as a percentage.

### 2.5. Determination of TNF-α, IL-1β, and IL-18

Cells in the logarithmic growth phase (5 × 10^4^ cells/mL) were seeded in 24-well plates, 1 mL per well. Different concentrations of polyphenols (the final concentrations were 20, 40, and 60 μg/mL, respectively) were added to all the PSLP groups and cultured for 6 h, and then MSU solution was added for further culture. After 24 h of culture, the cell supernatant was collected and centrifuged at 4 °C and 3000 r/min for 10 min. TNF-α, IL-1β, and IL-18 levels in the cell culture supernatant were determined according to the ELISA kit instructions.

### 2.6. Determination of ROS

The ROS of PSLP treatment on RAW264.7 cells according to the method described by Wan [17] with slightly modified. RAW264.7 cells were inoculated in 96-well plates, and after the above six groups of cells were cultured for 24 h, the supernatant was removed, the cells were washed once with serum-free DMEM medium, 100 µL of 10 µM DCFH-DA was added, incubated for 30 min at 37 °C in an incubator protected from light, and the cells were washed three times with serum-free DMEM to fully remove the unentered DCFH-DA, and observed under a fluorescence microscope. Analysis was performed using ImageJ 1.8.0 software.

### 2.7. Determination of SOD and MDA

The method for the detection of SOD and MDA was referenced from Cermeno [18] and Shen [19] with slightly modified. RAW264.7 cells were inoculated in 6-well plates, and the above six groups of cells were cultured for 24 h. After removing the supernatant, each group of cells was collected, and cell protein concentration was detected. The contents were calculated by assaying according to the instructions. The cytokine quantification of SOD and MDA was performed following the manufacturer’s specifications.

### 2.8. Determination of mRNA Expression of TNF-α, IL-1β, and IL-18 by qRT-PCR

The RNA was extracted from the above six groups of cells based on the manufacturer’s instructions, and the purity and concentration of RNA were determined at a ratio of 1.8–2.0 A_260_/A_280_. The cDNA was synthesized according to the instructions of the reverse transcription kit under the following reaction conditions: 42 °C for 2 min, 37 °C for 15 min, and 85 °C for 5 s. cDNA was stored at −20 °C. The primers are shown in Table 1.

### 2.9. RNA-Seq Analysis

Total RNA extraction and mRNA were enriched using magnetic beads with Oligo(dT) for cDNA library construction at Beijing Allwe Gene Technology Co., Ltd. (Beijing, China). Then, sequencing was performed by Illumina second-generation high-throughput sequencing platform using the PE150 sequencing strategy. The raw reads obtained from Illumina sequencing (FASTQ format) were then filtered to obtain clean reads. According to methods [20], DEseq (version 1.10.1) was used to identify the differentially expressed genes (DEGs) with | log_2_(fold change)| > 1 and *p* < 0.05, and that R (version 3.3.3) was then used for principal component analysis (PCA), volcano mapping, and clustering analysis. Meanwhile, the hypergeometric test was applied to perform significant enrichment analysis on pathways to identify those with significant enrichment of DEGs by the Kyoto Encyclopedia of Genes and Genomes (KEGG) pathway.

### 2.10. Data Analysis

Statistical analysis was performed by Prism 9 (GraphPad software, San Diego, CA, USA). Data are presented as the mean ± SD and were compared between the treatment and control groups using one-way ANOVA and *t*-tests. A *p*-value < 0.05 was considered to indicate statistical significance.

## 3. Results

### 3.1. Effect of PSLP on the Viability of RAW264.7 Cells

The effects of different concentrations of PSLP on RAW264.7 cell viability at 24 h and 48 h are shown in Figure 1. Compared to the MSU group, the viability of RAW 264.7 cells treated with 20 μg/mL and 40 μg/mL PSLP showed no obvious change. The viability of RAW 264.7 cells treated with 60 μg/mL PSLP showed an obvious change. The results showed that 20–60 μg/mL of PSLP were not cytotoxic to RAW264.7 cells. So 20–60 μg/mL of PSLP were selected for subsequent studies.

### 3.2. Effect of PSLP on Key Factors of Inflammation in RAW264.7 Cell

#### 3.2.1. IL-1β Secretion and Its mRNA Expression Level

IL-1β, also known as leukocyte pyrogen, is mainly produced by precursor cells, such as monocytes and macrophages, in the innate immune system. It has strong pro-inflammatory activity and induces the release of various pro-inflammatory mediators (TNF-α, IL-6), thereby promoting the activation of lymphocytes and immune cells and further amplifying inflammation [21]. Therefore, the detection of IL-1β levels in cells can also indirectly reflect the degree of cellular inflammation. As shown in Figure 2A, the release of the inflammatory factor IL-1β in the cell culture medium was highly significant (*p* < 0.01) after MSU stimulation of RAW264.7 cells. Compared with the MSU group, the levels of IL-1β in the cell cultures of the 20, 40, and 60 μg/mL PSLP group were highly significantly reduced (*p* < 0.01), and the levels of IL-1β in the cell cultures of the positive control group were highly significantly reduced (*p* < 0.01) but did not reach the level of the control group. The results suggest that MSU can induce the inflammatory response in RAW264.7 cells and that PSLP can significantly alleviate the inflammatory response induced by MSU in RAW264.7 cells.

On the other hand, it can be seen from Figure 2B that, compared with the control group, the mRNA expression level of IL-1β was significantly increased after MSU stimulation of RAW264.7 cells (*p* < 0.01). Furthermore, compared with the MSU group, 40 and 60 μg/mL of PSLP could significantly down-regulate the expression of IL-1β mRNA (*p* < 0.01), especially the 60 μg/mL PSLP group, which was close to the positive control group, while the 20 μg/mL PSLP group had no significant difference. This finding indicates that the higher the PSLP concentration, the lower the mRNA expression level of IL-1β in RAW264.7 cells.

#### 3.2.2. IL-18 Secretion and Its mRNA Expression Level

IL-18, also known as an interferon-γ inducing factor, is an early warning molecule similar to its family member IL-1β [22]. In Figure 3A, compared with the control group, the release of the inflammatory factor IL-18 in the cell culture medium of the model group was significantly increased after MSU stimulated RAW264.7 cells (*p* < 0.01). Compared with the MSU group, the release of IL-18 in the cell culture medium of the 40 and 60 μg/mL PSLP groups was significantly decreased (*p* < 0.01). It was worth noting that the release of IL-18 in the 60 μg/mL PSLP group was close to that in the positive control group treated with colchicine, and there was no significant difference between the 60 μg/mL PSLP group and the control group. Those results suggested that MSU could induce an inflammatory response in RAW264.7 cells and that PSLP could significantly inhibit the level of IL-18 secretion from RAW264.7 cells induced by MSU.

The mRNA expression levels of IL-18 were similar to those of IL-1β (Figure 4B). After MSU stimulation of RAW264.7 cells, the expression levels of IL-18 mRNA in the MSU group were significantly higher than those in the control group (*p* < 0.01). Compared with the MSU group, the 20, 40, and 60 μg/mL PSLP and colchicine treatment groups significantly down-regulated the expression of IL-18 mRNA (*p* < 0.01).

#### 3.2.3. TNF-α Secretion and Its mRNA Expression Level

TNF-α is one of the important inflammatory factors in gouty arthritis. It stimulates the production of IL-6, IL-8, and other inflammatory factors, promote the body ‘s inflammatory response and impels the transformation of monocytes into macrophages [23]. As displayed in Figure 4A, the release of inflammatory factor TNF-α in the cell culture medium was highly significant (*p* < 0.01) after MSU stimulation of RAW264.7 cells in the MSU group compared to the control group, indicating that MSU was able to establish an inflammatory cell model by inducing the development of inflammatory response in RAW264.7 cells. Compared with the MSU group, the content of TNF-α in the cell culture medium was significantly lower (*p* < 0.01) in the group with 40 and 60 μg/mL of PSLP, and the content of TNF-α in the colchicine group was significantly lower (*p* < 0.01). The results indicated that MSU was able to induce excessive TNF-α production by RAW264.7 cells, thus promoting the inflammatory response, while PSLP significantly inhibited TNF-α secretion by MSU-induced RAW264.7 cells.

Figure 4B shows the mRNA expression of TNF-α in RAW264.7 cells in all groups. The mRNA expression of TNF-α in RAW264.7 cells in the MSU group was 2.25 times higher than that in the control group. The inhibition rate of each concentration of PSLP on TNF-α mRNA expression ranged from 17.07–60.15%. PSLP treatment had a strong inhibitory effect on TNF-α gene expression. In summary, plum polyphenols can inhibit the expression of cellular inflammatory factors at the protein expression and gene transcription levels, reduce the body‘s inflammatory response, and thus play an anti-inflammatory role.

#### 3.2.4. Effect of PSLP on the Antioxidant Activity

As shown in Figure 5 and Figure 6, compared with the control group, the levels of ROS and MDA in the MSU group were significantly increased (*p* < 0.01), and the levels of SOD were significantly decreased (*p* < 0.01). After adding 20, 40, and 60 μg/mL PSLP, the expression of ROS and MDA decreased significantly (*p* < 0.01), and the levels of SOD increased. Compared with the MSU group, the antioxidant effect of 60 μg/mL PSLP was the most obvious (*p* < 0.01), which was close to the experimental effect of colchicine. From the fluorescence image in Figure 6B, the ROS production levels of RAW264.7 macrophages in the MSU treatment group were significantly higher than those in the control group, while the ROS production levels in the 20, 40, and 60 μg/mL polyphenol groups and the colchicine group were significantly lower than those in the MSU treatment group. The results revealed that MSU promoted the oxidative stress response of RAW264.7 macrophages and that PSLP could significantly inhibit MSU-induced oxidative stress.

### 3.3. Transcriptome Analysis in MSU-Induced RAW264.7 Cells

#### 3.3.1. Differentially Expressed mRNAs

To understand the signaling pathway of the anti-inflammatory mechanism of PSLP on RAW264.7 cells, we performed a transcriptome analysis using RNA-seq. The quality of the sequencing data of nine samples of RAW264.7 cells was evaluated. A total of 205018120 original sequencing data were obtained in this experiment. The clean reads were used for subsequent analysis, and the amount of offline data for each library sample sequenced was no less than 6G. The data summary is shown in Table 2. The proportion of bases with base mass values greater than 20 (Q20) and 30 (Q30) in the total bases of the 9 groups of samples was greater than 96% and 90%, respectively, and the error rate was 0.03%, indicating that the sequencing results were good.

The clean data of nine RAW264.7 macrophage samples were aligned to the reference genome using STAR alignment software, and 96.72% of 202971150 reads could be aligned to the mouse genome, with the majority (46.61%) of sequenced fragments located in exonic regions. The RNA-Seq correlation check is shown in Figure 7A. The correlation coefficients for the two parallel groups within the same group were all > 0.982, demonstrating that the samples were well parallel. All samples were subjected to PCA analysis based on gene expression, and the results revealed that PSLP treatment had an impact on gene expression in RAW264.7 cells. The sample correlation test‘s findings were supported by the experiment‘s strong biological replicates and the close proximity of samples within the group.

In order to analyze the related genes regulated by PSLP in RAW264.7 macrophages, the differentially expressed genes were summarized in a transcriptome analysis. The genes with significant differences in expression in the samples were screened by DESeq (1.10.1), and | log_2_ (FoldChange) | > 2 and qvalue < 0.05 were used as screening criteria to determine the significance of gene expression. The results of Figure 7 C,D showed that there were 8585 differentially expressed genes between the MSU group and the control group, including 9699 up-regulated genes and 5678 down-regulated genes. Compared with the MUS group, 8585 genes were differentially expressed in the polyphenol group, including 4421 up-regulated genes and 4164 down-regulated genes. Compared with the control group, there were 11,907 differentially expressed genes in the polyphenol group, including 7012 up-regulated genes and 4895 down-regulated genes.

#### 3.3.2. GO Enrichment Analysis

Gene Ontology (GO) is a standard gene function classification database with dynamic updates to characterize genes and their products in biological species [24]. A GO function analysis was performed on the differentially expressed genes in the polyphenol group and the MSU group. These differentially expressed genes were classified into three categories: biological processes, cellular components, and molecular function in functional prediction. The top 30 most significant GO terms were displayed in a histogram (Figure 8). Among the 30 most significant GO items in the polyphenol group and the MSU group, the most items in the total annotation amount were biological processes (16 items), and most genes were mainly enriched in organic substance metabolic process, cellular metabolic process, etc.; followed by cell component categories (12 items), most of which are mainly related to intracellular components, organelles, intracellular organelles, and membrane-bound organelles; the category of molecular function accounted for less total annotation (2 items), and most of the genes were mainly related to binding function and protein binding. This indicates that PSLP may be involved in the regulation of the development of RAW264.7 macrophages through the above biological functions.

#### 3.3.3. KEGG Pathway Enrichment Analysis

The top 20 pathways with the highest DEG enrichment in the polyphenol group compared to the MSU group were visible in the enrichment bubble plots of candidate genes (Figure 9). The enrichment degree of KEGG is measured by the enrichment factor, *p*-value, and the number of genes enriched in the pathway. The larger the enrichment coefficient, the smaller the *p*-value, and the more important the metabolic pathway [25]. The HIF-1 signaling pathway was the most enriched, followed by renal cell carcinoma and the ErbB signaling pathway. The anti-inflammatory effect of PSLP in RAW264.7 macrophages is mainly involved in the following signaling pathways, including the HIF-1 signaling pathway, the FoxO signaling pathway, the ErbB signaling pathway, ubiquitin-mediated proteolysis, tight junction, renal cell carcinoma, proteoglycan in cancer, PD-L1 expression and PD-1 checkpoint in cancer, amino acid biosynthesis, etc. Among them were the most differentially expressed genes involved in the herpes simplex virus 1 infection pathway, a total of 236.

## 4. Discussion

Gout is a chronic inflammatory disease caused by disorders of purine metabolism or reduced uric acid excretion [26]. AGA is the most common initial symptom of gout, and MSU-induced inflammation is the core pathological feature of gout. The mononuclear macrophage system plays a central role in the initiation, progression, and remission of MSU-triggered gout inflammation. MSU can directly stimulate monocytes/macrophages to regulate the release of inflammatory factors such as TNF-α, IL-1β, and IL-18, thereby promoting gout inflammatory response. In this study, RAW264.7 macrophages were stimulated with MSU, and it was found that the contents of TNF-α, IL-1β, and IL-18 in the supernatant and their mRNA expression in the cells were significantly increased. These indicate that MSU can induce a RAW264.7 macrophage inflammatory response, and a gout inflammatory cell model was successfully established.

Oxidative stress injury is mainly caused by reactive oxygen species. When the body is disturbed by external stimuli, it will accelerate the formation of ROS and aggravate lipid oxidation. MDA is a product of lipid peroxidation. The production of a large amount of ROS and MDA will indirectly reflect the degree of cell damage. SOD is an important part of the antioxidant defense system, which can remove excessive ROS and MDA in the body and protect cells and tissues [27]. The level of SOD activity indirectly reflects the body‘s ability to resist oxidative stress. This study confirmed that MSU could significantly increase the content of ROS and MDA in RAW264.7 macrophages and significantly reduce the content of SOD, indicating that MSU can promote the oxidative stress response of RAW264.7 macrophages.

PSLP is a natural compound from the fruit of the plum tree. Previously, we found that plum polyphenols can inhibit xanthine oxidase, thereby blocking the formation of uric acid in the body, and therefore have the potential to prevent gout caused by elevated serum uric acid. However, the effect and specific mechanism of PSLP on gout inflammation have not been elucidated. In this study, RAW264.7 cells were treated with different concentrations of PSLP. It was found that PSLP could significantly inhibit the secretion of pro-inflammatory factors TNF-α, IL-1β, and IL-18 in RAW264.7 macrophages and down-regulate their mRNA expression in cells. In addition, PSLP could increase the expression of SOD in RAW264.7 macrophages and inhibit the levels of ROS and MDA in cells, thus alleviating cell inflammation. The anti-inflammatory effect of PSLP had a certain dose-effect relationship, and the effect of 60 μg/mL PSLP was the most obvious. These results suggested that PSLP could inhibit MSU-induced cellular inflammatory responses and oxidative stress.

RNA-seq has been a new technology in recent years. It can simultaneously analyze the expression of thousands of genes at the transcriptional level in the same or different specimens and is considered to be a high-throughput, rapid, effective, high-resolution, and high-sensitivity gene expression analysis method [28,29]. Therefore, RNA-seq plays an important role in the study of cell physiological activities and biological metabolic mechanisms. In this study, transcriptome sequencing techniques and analysis were used to analyze the gene expression of RAW264.7 macrophages induced by MSU under the action of PSLP. The sequencing results showed that the number of genes with significantly up-regulated expression (4421) was slightly higher than that of down-regulated genes (4164) in RAW264.7 cells under the action of PSLP, suggesting that its promoting effect on cells was slightly stronger than its inhibitory effect. The results of GO functional classification analysis observed that the total annotations of biological process categories in the main functional prediction were the most in the three major categories, mainly enriched in organic matter metabolism processes, cell metabolism processes, and so on. The second was the cell component, mainly involving cell composition, organelles, intracellular organelles, membrane-bound organelles, etc. Most genes in the molecular function category were associated with binding function and protein binding. KEGG analysis exhibited that the main signaling pathways involved in the anti-inflammatory effect of PSLP in RAW264.7 macrophages include the HIF-1 signaling pathway, the FoxO signaling pathway, and the ErbB signaling pathway related to the occurrence and development of macrophage inflammation. The most differentially expressed genes involved in the herpes simplex virus type 1 infection pathway were 236.

Hypoxia-inducible factor1(HIF-1) plays a key role in cellular adaptation to altered oxygen supply and can act as a transcription factor to alter gene expression [30]. HIF-1α is a functional subunit of HIF-1 that mainly regulates the hypoxic cellular response, and its expression is regulated by oxygen at the protein level [31]. In addition to regulating cellular immune metabolism, HIF-1α may be involved in the regulation of cellular hypoxia. In addition to regulating cellular immune metabolism, it may also be involved in the regulation of cellular autophagy. HIF-1 α also affects the recruitment, migration, phagocytosis, killing, and other functions of immune cells and plays an important role in the occurrence and repression of inflammation [32]. Hydroxylase inhibitors can exert their anti-inflammatory [33] effects by activating the HIF-1 α pathway, and they have been applied in multiple inflammatory disease models. By inhibiting ROS, it also reduced the stability of HIF-1 α [34]. KEGG analysis exhibited that the HIF-1 signaling pathway was the most enriched pathway in the regulation of MSU-induced RAW264.7 macrophage inflammation by PSLP. In addition, there were 87 differentially expressed genes (DEGs), of which 34 genes were down-regulated, and 53 genes were up-regulated. It is further suggested that PSLP may regulate the secretion of ROS and SOD by inhibiting the activation of the HIF-1 signaling pathway to alleviate uric acid-induced inflammation.

Forkhead box transcription factor O1 (FoxO1) is a transcription factor in the nucleus that acts as a monomer and binds to its associated DNA target sequences to regulate downstream targets such as Bcl-2, Fas ligand, p27, phosphatase, and cell cycle proteins. Their effects mainly include inhibition of cell proliferation, promotion of apoptosis, and cell cycle arrest [35]. The activation of PI3K/AKT, a fundamental signaling pathway in the organism, phosphorylates the FoxO1 protein, and the phosphorylated FoxO1 undergoes nuclear translocation, resulting in the inability of FoxO1 to act on its target genes and the loss of its transcription factor role [36]. Liu [37] showed that tanshinone IIA could regulate biological processes such as inflammation, oxidative stress, and apoptosis in mouse cells by inhibiting the PI3K/AKT/FoxO1 pathways. Cui [38] also showed that dexmedetomidine reduced the level of inflammation, oxidative stress, and thus lipopolysaccharide-induced acute lung injury by inhibiting the PI3K/AKT/FoxO signaling pathways. KEGG analysis also showed the presence of a total of 83 DEGs in the FoxO signaling pathway map, of which 34 genes were down-regulated and 49 genes were up-regulated. It is further suggested that PSLP may regulate the secretion of TNF-α, IL-18, and IL-1β by inhibiting the activation of the PI3K/AKT/FoxO signaling pathways to alleviate MSU-induced inflammation when the organism enters a stressful state. The HIF-1 signaling pathway and the FoxO signaling pathway are activated, which are transferred from the cytoplasm to the nucleus, prompting the transcription of DEGs and up-regulating the expression of inflammatory factors. The results of this study showed that, compared with the MSU group, PSLP exerted anti-oxidative stress effects by blocking the HIF-1 signaling pathway and the FoxO signaling pathway.

## 5. Conclusions

The findings of the present study suggested that PSLP had significant anti-inflammatory activity at the cellular level. By activating the SOD activity of RAW264.7 macrophages and inhibiting the expression of inflammatory factors such as TNF-α, IL-1β, and IL-18 in cells, PSLP could effectively reduce the levels of ROS and MDA in cells and alleviate cell inflammation. The anti-inflammatory effect of plum polyphenols had a certain dose-effect relationship, and the effect of 60 ug/mL PSLP was the most obvious. 

The study followed the differential gene screening requirements by RNA-seq and revealed 8585 DEGs between the PSLP-treated group and the MUS group. From GO analysis, PSLP could affect the occurrence and development of RAW264.7 macrophage inflammation through biological processes, such as organic substance metabolism, intracellular organelles, and binding function. The regulation mechanism of PSLP on MSU-induced RAW264.7 macrophage inflammation may be achieved through the HIF-1 signaling pathway, renal cell carcinoma, the ErbB signaling pathway, and the FoxO signaling pathway. 

Nevertheless, these findings of cell cultures cannot be directly applied to in vivo conditions, so further studies are needed to explore the bioactivity, metabolism, tissue distribution, and excretion mechanisms of PSLP in using inflammatory animal models to confirm our findings.

## Figures and Tables

**Figure 1 foods-12-00254-f001:**
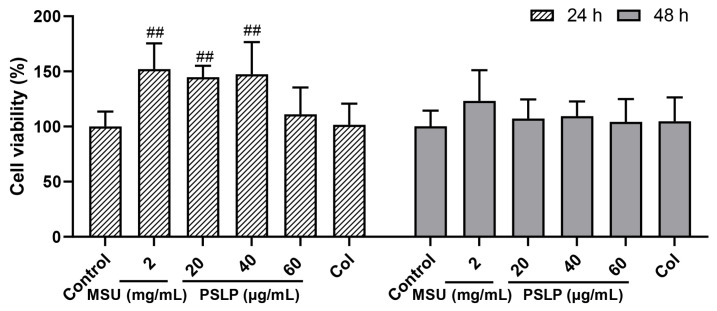
Effect of PSLP on the viability of RAW264.7 cells. The values were expressed as means ± SD (*n* = 3), ^##^
*p* < 0.01 vs. the MSU group.

**Figure 2 foods-12-00254-f002:**
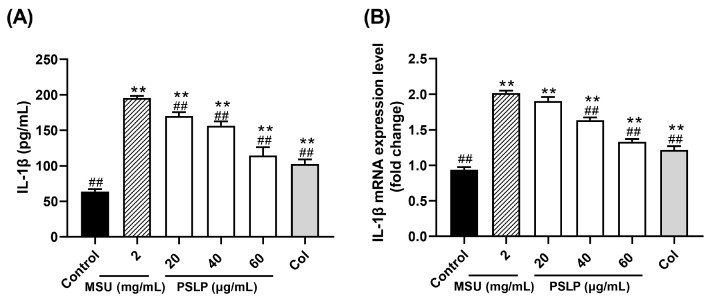
Effect of PSLP on IL-1β secretion (**A**) and its mRNA expression level (**B**) in MSU-stimulated RAW264.7 cells. The values were expressed as means ± SD (*n* = 3), ** *p* < 0.01 vs. the control group. ^##^
*p* < 0.01 vs. the MSU group.

**Figure 3 foods-12-00254-f003:**
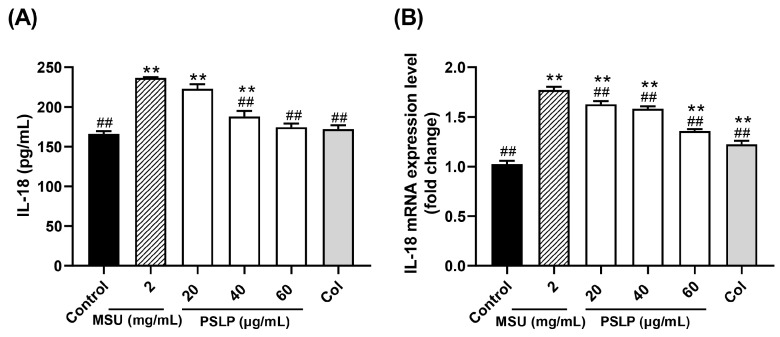
Effect of PSLP on IL-18 secretion (**A**) and its mRNA expression level (**B**) in MSU-stimulated RAW264.7 cells. The values were expressed as means ± SD (*n* = 3), ** *p* < 0.01 vs. the control group. ^##^
*p* < 0.01 vs. the MSU group.

**Figure 4 foods-12-00254-f004:**
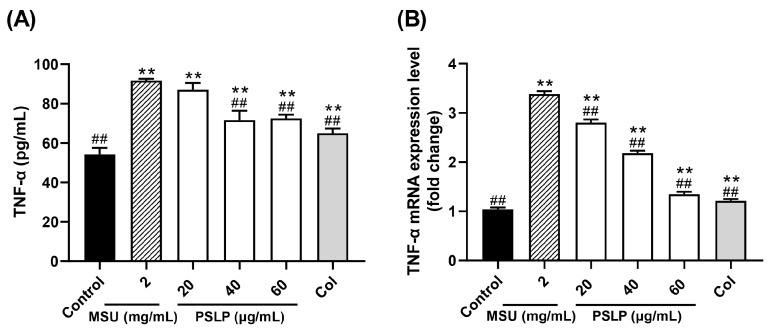
Effect of PSLP on TNF-α secretion (**A**) and its mRNA expression level (**B**) in MSU-stimulated RAW264.7 cells. The values were expressed as means ± SD (*n* = 3), ** *p* < 0.01 vs. the control group. ^##^
*p* < 0.01 vs. the MSU group.

**Figure 5 foods-12-00254-f005:**
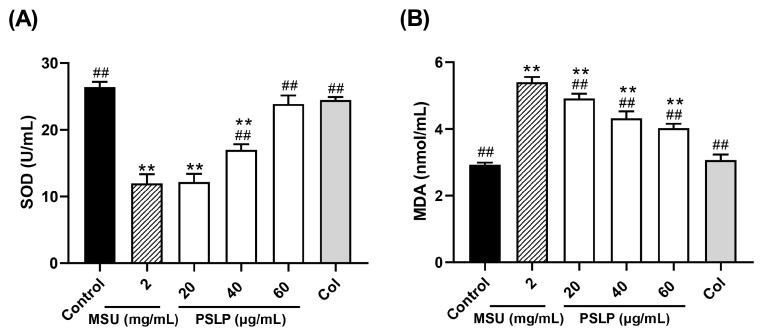
Effect of PSLP on intracellular SOD (**A**) and MDA (**B**) levels in MSU-stimulated RAW264.7 cells. The values were expressed as means ± SD (*n* = 3), ** *p* < 0.01 vs. the control group. ^##^
*p* < 0.01 vs. the MSU group.

**Figure 6 foods-12-00254-f006:**
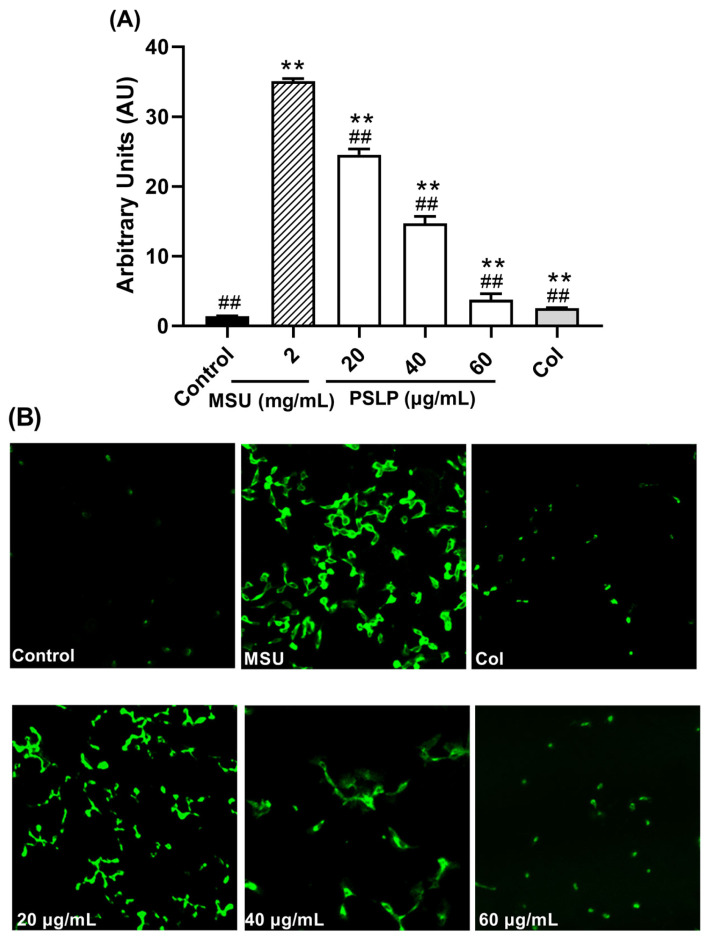
Effect of PSLP on intracellular ROS levels in MSU-stimulated RAW264.7 cells (**A**); ROS fluorogram of RAW264.7 cells (**B**). The values were expressed as means ± SD (*n* = 3), ** *p* < 0.01 vs. the control group. ^##^
*p* < 0.01 vs. the MSU group.

**Figure 7 foods-12-00254-f007:**
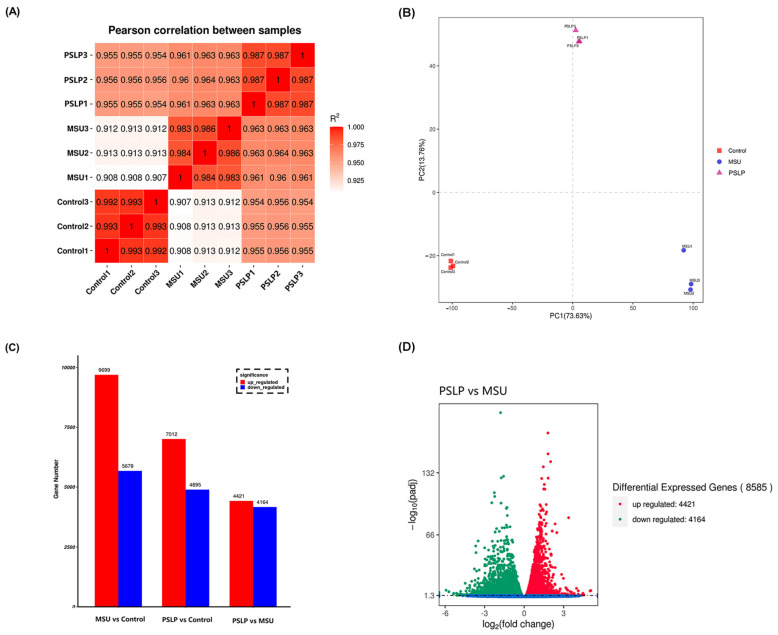
The changes in mRNA expression profiles in PSLP-treated RAW264.7 cells. (**A**) A sample correlation test. The darker red color represents the stronger correlation between groups. (**B**) PCA analysis of the whole transcriptomes of macrophages in the MSU group, PSLP group, and Control group. Different colors in the graph represent different groups. (**C**) Statistics of DEGs from the comparison of different treatment groups. (**D**) A volcano plot of DEGs between different treatment groups.

**Figure 8 foods-12-00254-f008:**
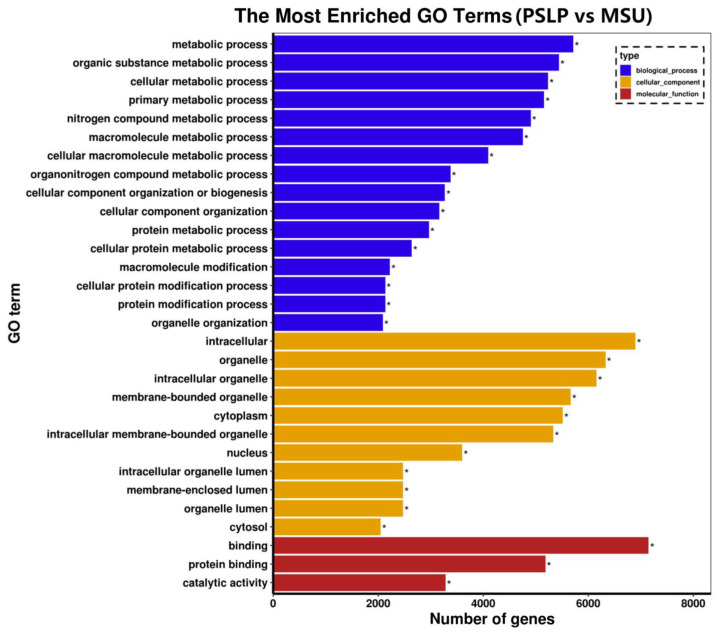
The GO enrichment analysis of the DEGs. The vertical coordinate is the enriched GO term, and the horizontal coordinate is the number of DEGs in the term. Different colors are used to distinguish between biological processes, cellular components, and molecular functions, with “*” indicating the significantly enriched GO terms.

**Figure 9 foods-12-00254-f009:**
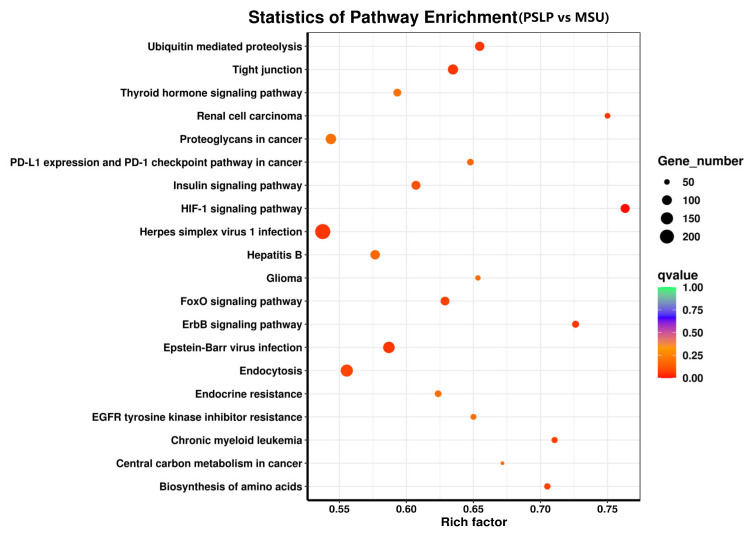
The KEGG pathway enrichment analysis of DEGs. KEGG pathway analysis of DEGs comparing the PSLP group and the MSU group. The dot size indicates the amount of DEG enriched in the pathway. The darker the color, the more significant the enrichment result.

**Table 1 foods-12-00254-t001:** Fluorescent quantitative PCR primer sequences.

Genes	Upstream Primer (5′→3′)	Downstream Primer (5′→3′)
*TNF-α* *IL-1β*	AGGCACTCCCCCAAAAGATG	CCACTTGGTGGTTTGTGAGTG
	TGGGTACTGGAGAGTGGTCA	GGCTTGGGAGTGAAGAGGTC
*IL-18*	CCAGTGGCTGCAGATTCAGA	CTCTGCTTCGGTCCCAACAT

**Table 2 foods-12-00254-t002:** Fluorescent quantitative PCR primer sequences.

Sample Name	Raw Reads	Raw Bases	Clean Reads	Clean Bases	Error Rate	Q20	Q30	GC Content
Control1	44996622	6.74G	44533488	6.68G	0.03%	97.08%	91.25%	52.14%
Control2	48495526	7.27G	48003574	7.20G	0.03%	97.01%	91.14%	52.17%
Control3	41830938	6.27G	41386724	6.21G	0.03%	97.05%	91.17%	52.16%
MSU1	41584164	6.23G	41251536	6.19G	0.03%	96.90%	90.92%	51.24%
MSU2	55093556	8.26G	54567276	8.19G	0.03%	96.91%	90.92%	51.52%
MSU3	46341452	6.95G	45927324	6.89G	0.03%	97.08%	91.33%	51.43%
PSLP1	42896162	6.43G	42483810	6.37G	0.03%	97.23%	91.69%	51.41%
PSLP2	46681628	7.00G	46047436	6.91G	0.03%	96.90%	90.95%	51.77%
PSLP3	42116192	6.31G	41741132	6.26G	0.03%	96.66%	90.32%	51.60%

## Data Availability

The data of transcriptomics have been deposited in NCBI SRA under the accession number PRJNA916348. Other data is contained within the article.

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
