# Peer review of "Anti-Inflammatory Effects of Polyphenols from Plum (Prunus salicina Lindl) on RAW264.7 Macrophages Induced by Monosodium Urate and Potential Mechanisms"

_foods, 2023, doi:10.3390/foods12020254_

Round 1
Reviewer 1 Report
The manuscript titled: “Anti-inflammatory effects of polyphenols from plum (Prunus salicina Lindl.) on RAW264.7 macrophages induced by mono-sodium urate and potential mechanisms” is well written. The manuscript is based on a well-constructed scientific concept and carried out the studies are well. However, the functional studies of macrophages are missing in the manuscript. The present manuscript would be benefited by addressing the points below. I would suggest minor revision.
Comments:
· Authors have used the extracted polyphenols from plum; authors need to characterize the sample using the HPLC used in the studies.
· Authors must show the endotoxin levels in the extracted polyphenol sample used in the studies.
· Authors showed that Polyphenols regulate the proinflammatory phenotypes of macrophages; however, the manuscript lacks the macrophage functionality data, such as phagocytosis, and the effect of Polyphenolics and quantification of endotoxin in the sample; below paper will be helpful to this manuscript; kindly refer it. “Prasanna VK, Venkatesh YP. Characterization of onion lectin (Allium cepa agglutinin) as an immunomodulatory protein inducing Th1-type immune response in vitro. Int Immunopharmacol. 2015 Jun;26(2):304-13. doi: 10.1016/j.intimp.2015.04.009.”
Author Response
Thank you for your valuable comments, my response is attached.

Reviewer 2 Report
To the Authors of the manuscript ‘Anti-inflammatory effects of polyphenols from plum (Prunus salicina Lindl.) on RAW264.7 macrophages induced by mono-sodium urate and potential mechanisms’.
I have thoroughly reviewed the subject matter of the manuscript and I am impressed. The work has been carried out with care, the presented results are significant and, in my judgment, sufficient to meet the interest for broad readership. However, I regret to say that there are some minor examples of negligence. I pointed out my observations below.
- I have doubt if the manuscript is presented in a well-structured manner… The main reason is MSU-induced RAW264.7 macrophage inflammation model. In my opinion, the paragraph 2.3. Cell culture needs to be improved, the information about the times of treatment with MSU and then PSLP needs to be completed. For one can get the impression that part of the cultures were treated with MSU and the rest with PSLP at different concentrations... And this is not true, is it?
- The issue mentioned above has also consequences in the questionable graphical presentation of the results. Looking at the figures, I have doubts whether PSLP was intended to show biological activity in previously MSU-induced cells or whether the effects of the two substances are being compared with each other. Thus the figures are not easy to interpret and understand.
- PSLP was tested at the concentration: 20, 40, and 60 µg/mL. My question is why lower and especially higher concentrations were not consulted? A comment or even the presentattion of preliminary studies are required.
- In the lines 88-91, the parameters of an ultrasonic bath are named. Please clarify wheter 300 W or 420 W was used during the extraction of polyphenols.
- In the line 109, I suggest CO2 instead of CO2.
- The paragraph 2.5. Determination of TNF (…): what do you mean as ‘the drug groups’? Please, explain.
- In the lines 137-138, I suggest some reorganisation such as: ‘100 µL of 10 µM DCFH-DA …’. There are also other similar examples of misplaced provisions in the manuscript that need to be corrected.
- In the figures 2b, 3b and 4b, I suggest changing the title of the y-axis. It should refer to the information that these are the results of mRNA expression levels. Otherwise, figure b may be confused by the reader with figure a, merely illustrating a different record of the results.
- The line 266: Please explain, what do you mean as ‘the blank group’
- The quality of Figure 7 needs to be improved.
- I recommend to consult the manuscript with a natvie speaker.
- Editorial corrections are required in the whole text (pauses, points, comas, italic script).
- Please check the use of abbreviations.
Author Response

(The authors gave the same response as above.)
